# Optimizing Instructional Policies

**Robert V. Lindsey**[*], **Michael C. Mozer**[*], **William J. Huggins**[*], **Harold Pashler**[†]
[*] Department of Computer Science, University of Colorado, Boulder
[†] Department of Psychology, University of California, San Diego

## Abstract

Psychologists are interested in developing instructional policies that boost student learning. An instructional policy specifies the manner and content of instruction. For example, in the domain of concept learning, a policy might specify the nature of exemplars chosen over a training sequence. Traditional psychological studies compare several hand-selected policies, e.g., contrasting a policy that selects only difficult-to-classify exemplars with a policy that gradually progresses over the training sequence from easy exemplars to more difficult (known as *fading*). We propose an alternative to the traditional methodology in which we define a parameterized space of policies and search this space to identify the optimal policy. For example, in concept learning, policies might be described by a fading function that specifies exemplar difficulty over time. We propose an experimental technique for searching policy spaces using Gaussian process surrogate-based optimization and a generative model of student performance. Instead of evaluating a few experimental conditions each with many human subjects, as the traditional methodology does, our technique evaluates many experimental conditions each with a few subjects. Even though individual subjects provide only a noisy estimate of the population mean, the optimization method allows us to determine the shape of the policy space and to identify the global optimum, and is as efficient in its subject budget as a traditional A-B comparison. We evaluate the method via two behavioral studies, and suggest that the method has broad applicability to optimization problems involving humans outside the educational arena.

## 1   Introduction

What makes a teacher effective? A critical factor is their *instructional policy*, which specifies the manner and content of instruction. Electronic tutoring systems have been constructed that implement domain-specific instructional policies (e.g., J. R. Anderson, Conrad, & Corbett, 1989; Koedinger & Corbett, 2006; Martin & VanLehn, 1995). A tutoring system decides at every point in a session whether to present some new material, provide a detailed example to illustrate a concept, pose new problems or questions, or lead the student step-by-step to discover an answer. Prior efforts have focused on higher cognitive domains (e.g., algebra) in which policies result from an expert-systems approach involving careful handcrafted analysis and design followed by iterative evaluation and refinement. As a complement to these efforts, we are interested in addressing fundamental questions in the design of instructional policies that pertain to basic cognitive skills.

Consider a concrete example: training individuals to discriminate between two perceptual or conceptual categories, such as determining whether mammogram x-ray images are negative or positive for an abnormality. In training from examples, should the instructor tend to alternate between categories—as in PNPNPNPN for positive and negative examples—or present a series of instances from the same category—PPPPNNNN (Goldstone & Steyvers,

2001)? Both of these strategies—*interleaving* and *blocking*, respectively—are adopted by human instructors (Khan, Zhu, & Mutlu, 2011). Reliable advantages between strategies has been observed (Kang & Pashler, 2011; Kornell & Bjork, 2008) and factors influencing the relative effectiveness of each have been explored (Carvalho & Goldstone, 2011).

Empirical evaluation of blocking and interleaving policies involves training a set of human subjects with a fixed-length sequence of exemplars drawn from one policy or the other. During training, exemplars are presented one at a time, and typically subjects are asked to guess the category label associated with the exemplar, after which they are told the correct label. Following training, mean classification accuracy is evaluated over a set of test exemplars. Such an experiment yields an intrinsically noisy evaluation of the two policies, limited by the number of subjects and inter-individual variability. Consequently, the goal of a typical psychological experiment is to find a statistically reliable difference between the training conditions, allowing the experimenter to conclude that one policy is superior.

Blocking and interleaving are but two points in a space of policies that could be parameterized by the probability, $\rho$, that the exemplar presented on trial $t + 1$ is drawn from the same category as the exemplar on trial $t$. Blocking and interleaving correspond to $\rho$ near 1 and 0, respectively. (There are many more interesting ways of constructing a policy space that includes blocking and interleaving, e.g., $\rho$ might vary with $t$ or with a student's running-average classification accuracy, but we will use the simple fixed-$\rho$ policy space for illustration.) Although one would ideally like to explore the policy space exhaustively, limits on the availability of experimental subjects and laboratory resources make it challenging to conduct studies evaluating more than a few candidate policies to the degree necessary to obtain statistically significant differences.

## 2 Optimizing an instructional policy

Our goal is to discover the *optimum* in policy space—the policy that maximizes mean accuracy or another measure of performance over a population of students. (We focus on optimizing for a population but later discuss how our approach might be used to address individual differences.) Our challenge is performing optimization on a budget: each subject tested imposes a time or financial cost. Evaluating a single policy with a degree of certainty requires testing many subjects to reduce sampling variance due to individual differences, factors outside of experimental control (e.g., alertness), and imprecise measurement obtained from brief evaluations and discrete (e.g., correct or incorrect) responses. Consequently, exhaustive search over the set of distinguishable policies is not feasible.

Past research on optimal teaching (Chi, VanLehn, Litman, & Jordan, 2011; Rafferty, Brunskill, Griffiths, & Shafto, 2011; Whitehill & Movellan, 2010) has investigated reinforcement learning and POMDP approaches. These approaches are intriguing but are not typically touted for their data efficiency. To avoid exceeding a subject budget, the flexibility of the POMDP framework demands additional bias, imposed via restrictions on the class of candidate policies and strong assumptions about the learner. The approach we will propose likewise requires specification of a constrained policy space, but does not make assumptions about the internal state of the learner or the temporal dynamics of learning. In contrast to POMDP approaches, the cognitive agnosticism of our approach allows it to be readily applied to arbitrary policy optimization problems. Direct optimization methods that accommodate noisy function evaluations have also been proposed, but experimentation with one such technique (E. J. Anderson & Ferris, 2001) convinced us that the method we propose here is orders of magnitude more efficient in its required subject budget.

Neither POMDP nor direct-optimization approaches models the policy space explicitly. In contrast, we propose an approach based on *function approximation*. From a function-approximation perspective, the goal is to determine the shape and optimum of the function that maps policies to performance—call this the *policy performance function* or *PPF*. What sort of experimental design should be used to approximate the PPF? Traditional experimental design—which aims to show a statistically reliable difference between two alternative policies—requires testing many subjects for each policy. However, if our goal is to determine the shape of the PPF, we may get better value from data collection by evaluating a large

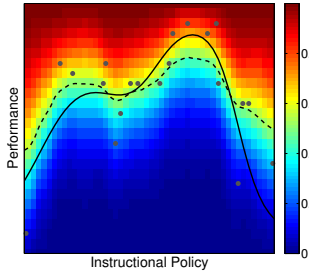

Figure 1: A hypothetical 1D instructional policy space. The solid black line represents an (unknown) policy performance function. The grey disks indicate the noisy outcome of single-subject experiments conducted at specified points in policy space. (The diameter of the disk represents the number of data points occuring at the disk's location.) The dashed black line depicts the GP posterior mean, and the coloring of each vertical strip represents the cumulative density function for the posterior.

number of points in policy space each with few subjects instead of a small number of points each with many subjects. This possibility suggests a new paradigm for experimental design in psychological science. Our vision is a completely automated system that selects points in policy space to evaluate, runs an *experiment*—an evaluation of some policy with one or a small number of subjects—and repeats until a budget for data collection is exhausted.

## 2.1 Surrogate-based optimization using Gaussian process regression

In surrogate-based optimization (e.g., Forrester & Keane, 2009), experimental observations serve to constrain a *surrogate* model that approximates the function being optimized. This surrogate is used both to select additional experiments to run and to estimate the optimum. Gaussian process regression (GPR) has long been used as the surrogate for solving low-dimensional stochastic optimization problems in engineering fields (Forrester & Keane, 2009; Sacks, Welch, Mitchell, & Wynn, 1989). Like other Bayesian models, GPR makes efficient use of limited data, which is particularly critical to us because our budget is expressed in terms of the number of subjects required. Further, GPR provides a principled approach to handling measurement uncertainty, which is a problem any experimental context but is particularly striking in human experimentation due to the range of factors influencing performance. The primary constraint imposed by the Gaussian Process prior—that of function smoothness—can readily be ensured with the appropriate design of policy spaces. To illustrate GPR in surrogate-based optimization, Figure 1 depicts a hypothetical 1D instructional policy space, along with the true PPF and the GPR posterior conditioned on the outcome of a set of single-subject experiments at various points in policy space.

## 2.2 Generative model of student performance

Each instructional policy is presumed to have an inherent effectiveness for a population of individuals. However, a policy's effectiveness can be observed only indirectly through measurements of subject performance such as the number of correct responses. To determine the most effective policy from noisy observations, we must specify a generative model of student performance which relates the inherent effectiveness of instruction to observed performance.

Formally, each subject $s$ is trained under a policy $\mathbf{x}_s$ and then tested to evaluate their performance. We posit that each training policy $\mathbf{x}$ has a latent population-wide effectiveness $f_{\mathbf{x}} \in \mathbb{R}$ and that how well a subject performs on the test is a noisy function of $f_{\mathbf{x}_s}$. We are interested in predicting the effectiveness of a policy $\mathbf{x}'$ across a population of students given the observed test scores of S subjects trained under the policies $\mathbf{x}_{1:S}$. Conceptually, this involves first inferring the effectiveness $\mathbf{f}$ of policies $\mathbf{x}_{1:S}$ from the noisy test data, then interpolating from $\mathbf{f}$ to $f_{\mathbf{x}'}$.

Using a standard Bayesian nonparametric approach, we place a mean-zero Gaussian Process prior over the function $f_{\mathbf{x}}$. For the finite set of S observations, this corresponds to the multivariate normal distribution $\mathbf{f} \sim \mathrm{MVN}(0, \Sigma)$, where $\Sigma$ is a covariance matrix prescribing how smoothly varying we expect $f$ to be across policies. We use the squared-exponential covariance function, so that $\Sigma_{s,s'} = \sigma^2 \exp(-\frac{||\mathbf{x}_s - \mathbf{x}_{s'}||^2}{2\ell^2})$, and $\sigma^2$ and $\ell$ as free parameters.

Having specified a prior over policy effectiveness, we turn to specifying a distribution over observable measures of subject learning conditioned on effectiveness. In this paper, we measure learning by administering a multiple-choice test to each subject $s$ and observing

the number of correct responses $s$ made, $c_s$, out of $n_s$ questions. We assume the probability that subject $s$ answers any question correctly is a random variable $\mu_s$ whose expected value is related to the policy's effectiveness via the logistic transform: $\mathbb{E}\left[\mu_s\right] = \mathrm{logistic}(o + f_{\mathbf{x}_s})$ where $o$ is a constant. This is consistent with the observation model

$$\mu_s \mid f_{\mathbf{x}_s}, o, \gamma \sim \mathrm{Beta}(\gamma, \gamma e^{-(o+f_{\mathbf{x}_s})}) \qquad c_s \mid \mu_s \sim \mathrm{Binomial}(g + (1-g)\mu_s;\ n_s) \qquad (1)$$

where $\gamma$ controls inter-subject variability in $\mu_s$ and $g$ is the probability of answering a question correctly by random guessing. In this paper, we assume $g = .5$. For this special case, the analytic marginalization over $\mu_s$ yields

$$P(c_s \mid f_{\mathbf{x}_s}, \gamma, o, g = .5) = 2^{-n_s} \binom{n_s}{c_s} \sum_{i=0}^{c_s} \binom{c_s}{i} \frac{\mathrm{B}(\gamma + i, n_s - c_s + \gamma e^{-(o+f_{\mathbf{x}_s})})}{\mathrm{B}(\gamma, \gamma e^{-(o+f_{\mathbf{x}_s})})} \qquad (2)$$

where $\mathrm{B}(a,b) = \Gamma(a)\Gamma(b)/\Gamma(a+b)$ is the beta function.

Parameters $\boldsymbol{\theta} \equiv \left\{\gamma, o, \sigma^2, \ell\right\}$ are given vague uniform priors. The effectiveness of a policy $\mathbf{x}'$ is estimated via $p(\mathbf{f}_{\mathbf{x}'} \mid \mathbf{c}) \approx \frac{1}{M}\sum_{m=1}^{M} p(\mathbf{f}_{\mathbf{x}'} \mid \mathbf{f}^{(m)}, \boldsymbol{\theta}^{(m)})$, where $p(\mathbf{f}_{\mathbf{x}'} \mid \mathbf{f}^{(m)}, \boldsymbol{\theta}^{(m)})$ is Gaussian with mean and variance determined by the $m$th sample from the posterior $p(\mathbf{f}, \boldsymbol{\theta} \mid \mathbf{c})$. Posterior samples are drawn via elliptical slice sampling, a technique well-suited for models with highly correlated latent Gaussian variables (Murray, Adams, & MacKay, 2010).

We have also explored a more general framework that relaxes the relationship between chance-guessing and test performance and allows for multiple policies to be evaluated per subject. With regard to the latter, subjects may undergo multiple randomly ordered blocks of trials where in each block $b$ a subject $s$ is trained under a policy $f_{\mathbf{x}_s^b}$ and then tested. The observation model is altered so that the score in a block is given by $c_s^b \sim \mathrm{Binomial}(\mu_s^b; n_s^b)$ where $\mu_s^b \equiv \mathrm{logistic}(o' + \alpha_s + f_{\mathbf{x}_s^b})$, the factor $\alpha_s \sim \mathrm{Normal}(0, \tau_\alpha^{-1})$ represents the ability of subject $s$ across blocks, and the constant $o'$ subsumes the role of $o$ and $g$ from the original model. In the spirit of item-response theory (Boeck & Wilson, 2004), the model could be extended further to include factors that represent the difficulty of individual test questions and interactions between subject ability and question difficulty.

## 2.3 Active selection

GP optimization requires a strategy for actively selecting the next experiment. (We refer to this as a 'strategy' instead of as a 'policy' to avoid confusion with instructional policies.) Many heuristic strategies have been proposed (Forrester & Keane, 2009), including: *grid sampling* over the policy space; expanding or contracting a *trust region*; and *goal-setting* approaches that identify regions of policy space where performance is likely to attain some target level or beat out the current best experiment result. In addition, greedy versus $k$-step predictive planning has been considered (Osborne, Garnett, & Roberts, 2009).

Every strategy faces an exploration/exploitation trade off. Exploration involves searching regions of the function with the maximum uncertainty; exploitation involves concentrating on the regions of the function that currently appear to be most promising. Each has a cost. A focus on exploration rapidly exhausts the subject budget for subjects. A focus on exploitation leads to selection of local optima.

The upper-confidence bound (UCB) strategy (Forrester & Keane, 2009; Srinivas, Krause, Kakade, & Seeger, 2010) attempts to avoid these two costs by starting in an exploratory mode and shifting to exploitation. This strategy chooses the most-promising experiment from an upper-confidence bound on the GPR: $\mathbf{x}_t = \mathrm{argmax}_{\mathbf{x}}\ \hat{\mu}_{t-1}(\mathbf{x}) + \eta_t \hat{\sigma}_{t-1}(\mathbf{x})$, where $t$ is a time index, $\hat{\mu}$ and $\hat{\sigma}$ are the mean and standard deviation of the GPR, and $\eta_t$ controls the exploration/exploitation trade off. Large $\eta_t$ focus on regions with the greatest uncertainty, but as $\eta_t \to 0$, the focus shifts to exploitation in the neighborhood of the current best policy. Annealing $\eta_t$ as a function of $t$ will yield exploration initially shifting toward exploitation.

We adapt the UCB strategy by transforming the UCB based on the GPR to an expression based on the the population accuracy (proportion correct) via $\mathbf{x}_t = \mathrm{argmax}_{\mathbf{x}} P(\frac{c_s}{n_s} > \nu_t \mid f_{\mathbf{x}})$, where $\nu_t$ is an accuracy level determining the exploration/exploitation trade off. In simulations, we found that setting $\nu_t = .999$ was effective. Note that in applying the UCB selection

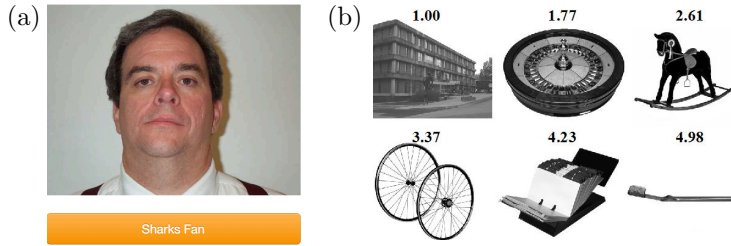

(a)                    (b)

Sharks Fan

Figure 2:
(a) Experiment 1 training display; (b) Selected Experiment 2 stimuli and their graspability ratings

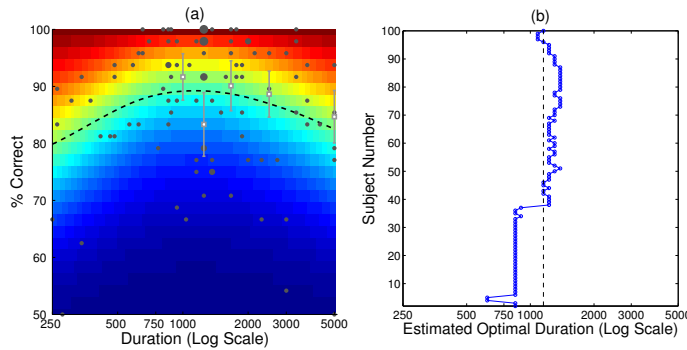

Figure 3: Experiment 1 results. (a) Posterior density of the PPF with 100 subjects. Light grey squares with error bars indicate the results of a traditional comparison among conditions. (b) Prediction of optimum presentation duration as more subjects are run; dashed line is asymptotic value.

strategy, we must search over a set of candidate policies. We applied a fine uniform grid search over policy space to perform this selection.

# 3   Experiment 1: Optimizing presentation rate

de Jonge, Tabbers, Pecher, and Zeelenberg (2012) studied the effect of presentation rate on word-pair learning. During training, each pair was viewed for a total of 16 sec. Viewing was divided into $16/d$ trials each with a duration of $d$ sec, where $d$ ranged from 1 sec (viewing the pair 16 times) to 16 sec (viewing the pair once). de Jong et al. found that an intermediate duration yielded better cued recall performance both immediately and following a delay.

We explored a variant of this experiment in which subjects were asked to learn the favorite sporting team of six individuals. During training, each individual's face was shown along with their favorite team—either Jets or Sharks (Figure 2a). The training policy specifies the duration $d$ of each face-team pair. Training was over a 30 second period, with a total of $30/d$ trials and an average of $5/d$ presentations per face-team pair. Presentation sequences were blocked, where a block consists of all six individuals in random order. Immediately following training, subjects were tested on each of the six faces in random order and were asked to select the corresponding team. The training/testing procedure was repeated for eight rounds each using different faces. In total, each subject responded to 48 faces. The faces were balanced across ethnicity, age, and gender (provided by Minear & Park, 2004).

Using Mechanical Turk, we recruited 100 subjects who were paid $0.30 for their participation. The policy space was defined to be in the logarithm of the duration, i.e., $d = e^x$, where $x \in [\ln(.25)\ \ln(5)]$. The space included only values of $x$ such that $30/d$ is an integer; i.e., we ensured that no trials were cut short by the 30 second time limit. Subject 1's training policy, $x_1$, was set to the median of the range of admissable values (857 ms). After each subject $t$ completed the experiment, the PPF posterior was reestimated, and the upper-confidence bound strategy was used to select the policy for subject $t + 1$, $x_{t+1}$.

Figure 3a shows the PPF posterior based on 100 subjects. (We include a movie showing the evolution of the PPF over subjects in the Supplementary Materials.) The diameter of the grey disks indicate the number of data points observed at that location in the space. The optimum of the PPF mean is at 1.15 sec, at which duration each face-team pair will be shown on expectation 4.33 times during training. Though the result seems intuitive, we've polled colleagues, and predictions for the peak ranged from below 1 sec to 2.5 sec. Figure 3b uses the PPF mean to estimate the optimum duration, and this duration is plotted against

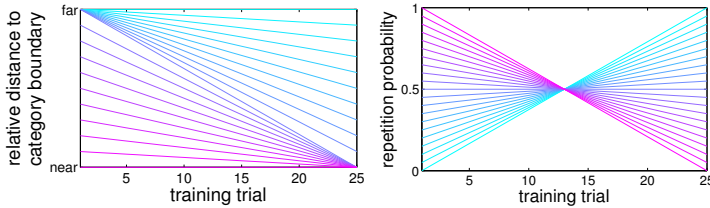

Figure 4: Expt. 2, trial dependent fading and repetition policies (left and right, respectively). Colored lines represent specific policies.

the number of subjects. Our procedure yields an estimate for the optimum duration that is quite stable after about 40 subjects.

Ideally, one would like to compare the PPF posterior to ground truth. However, obtaining ground truth requires a massive data collection effort. As an alternative, we contrast our result with a more traditional experimental study based on the same number of subjects. We ran 100 additional subjects in a standard experimental design involving evaluation of five alternative policies, $d \in \{1, 1.25, 1.667, 2.5, 5\}$, 20 subjects per policy. (These durations correspond to 1-5 presentations of each face-team pair during training.) The mean score for each policy is plotted in Figure 3a as light grey squares with bars indicating $\pm 2$ standard errors of the mean. The result of the traditional experiment is coarsely consistent with the PPF posterior, but the budget of 100 subjects places a limitation on the interpretability of the results. When matched on budget, the optimization procedure appears to produce results that are more interpretable and less sensitive to noise in the data. Note that we have biased this comparison in favor of the traditional design by restricting the exploration of the policy space to the region 1 sec $\leq d \leq$ 5 sec. Nonetheless, no clear pattern emerges in the shape of the PPF based on the outcome of the traditional design.

## 4    Experiment 2: Optimizing training example sequence

In Experiment 2, we study concept learning from examples. Subjects are told that martians will teach them the meaning of a martian adjective, GLOPNOR, by presenting a series of example objects, some of which have the property GLOPNOR and others do not. During a training phase, objects are presented one at a time and subjects must classify the object as GLOPNOR or NOT-GLOPNOR. They then receive feedback as to the correctness of their response. On each trial, the object from the previous trial is shown in the corner of the display along with its correct classification, the reason for which is to facilitate comparison and contrasting of objects. Following 25 training trials, 24 test trials are administered in which the subject makes a classification but receives no feedback. The training and test trials are roughly balanced in number of positive and negative examples.

The stimuli in this experiment are drawn from a set of 320 objects normed by Salmon, McMullen, and Filliter (2010) for *graspability*, i.e., how manipulable an object is according to how easy it is to grasp and use the object *with one hand*. They polled 57 individuals, each of whom rated each of the objects multiple times using a 1–5 scale, where 1 means not graspable and 5 means highly graspable. Figure 2b shows several objects and their ratings. We divided the objects into two groups by their mean rating, with the NOT-GLOPNOR group having ratings in [1, 2.75] and the GLOPNOR group having ratings in [3.25, 5]. (We discarded objects with ratings in [2.75, 3.25] because they are too difficult even if one knows the concept). The classification task is easy if one knows that the concept is graspability. However, the challenge of inferring the concept is extremely difficult because there are many dimensions along which these objects vary and any one—or more—could be the classification dimension(s).

We defined an instructional policy space characterized by two dimensions: *fading* and *blocking*. Fading refers to the notion from the animal learning literature that learning is facilitated by presenting exemplars far from the category boundary initially, and gradually transitioning toward more difficult exemplars over time. Exemplars far from the boundary may help individuals to attend to the dimension of interest; exemplars near the boundary may help individuals determine where the boundary lies (Pashler & Mozer, in press). Theorists have

also made computational arguments for the benefit of fading (Bengio, Louradour, Collobert, & Weston, 2009; Khan et al., 2011). Blocking refers to the issue discussed in the Introduction concerning the sequence of category labels: Should training exemplars be blocked or interleaved? That is, should the category label on one trial tend to be the same as or different than the label on the previous trial?

For fading, we considered a family of trial-dependent functions that specify the distance of the chosen exemplar to the category boundary (left panel of Figure 4). This family is parameterized by a single policy variable $x_2$, $0 \leq x_2 \leq 1$ that relates to the distance of an exemplar to the category boundary, $d$, as follows: $d(t, x_2) = \min(1, 2x_2) - (1 - |2x_2 - 1|)\frac{t-1}{T-1}$, where $T$ is the total number of training trials and $t$ is the current trial. For blocking, we also considered a family of trial-dependent functions that vary the probability of a category label repetition over trials (right panel of Figure 4). This family is parameterized by the policy variable $x_1$, $0 \leq x_1 \leq 1$, that relates to the probability of repeating the category label of the previous trial, $r$, as follows: $r(t, x_1) = x_1 + (1 - 2x_1)\frac{t-1}{T-1}$.

Figure 5a provides a visualization of sample training trial sequences for different points in the 2D policy space. Each graph represents an instance of a specific (probabilistic) policy. The abscissa of each graph is an index over the 25 training trials; the ordinate represents the category label and its distance from the category boundary. Policies in the top and bottom rows show sequences of all-easy and all-hard examples, respectively; intermediate rows achieve fading in various forms. Policies in the leftmost column begin training with many repetitions and end training with many alternations; policies in the rightmost column begin with alternations and end with repetitions; policies in the middle column have a time-invariant repetition probability of 0.5.

Regardless of the training sequence, the set of test objects was the same for all subjects. The test objects spanned the spectrum of distances from the category boundary. During test, subjects were required to make a forced choice GLOPNOR/NOT-GLOPNOR judgment.

We seeded the optimization process by running 10 subjects in each of four corners of policy space as well as in the center point of the space. We then ran 150 additional subjects using GP-based optimization. Figure 5 shows the PPF posterior mean over the 2D policy space, along with the selection in policy space of the 200 subjects. Contour map colors indicate the expected accuracy of the corresponding policy (in contrast to the earlier colored graphs in which the coloring indicates the cdf). The optimal policy is located at $\mathbf{x}^* = (1, .66)$.

To validate the outcome of this exploration, we ran 50 subjects at $\mathbf{x}^*$ as well as policies in the upper corners and the center of Figure 5. Consistent with the prediction of the PPF posterior, mean accuracy at $\mathbf{x}^*$ is 68.6%, compared to 60.9% for $(0, 1)$, 65.7% for $(1, 0)$, and 66.6% for $(.5, .5)$. Unfortunately, only one of the paired comparisons was statistically reliable by a two-tailed Bonferroni corrected $t$-test: $(0, 1)$ versus $\mathbf{x}^*$ ($p = .027$). However, post-hoc power computation revealed that with 50 subjects and the variability inherent in the data, the odds of observing a reliable 2% difference in the mean is only .10. Running an additional 50 subjects would raise the power to only .17. Thus, although we did not observe a statistically significant improvement at the inferred optimum compared to sensible alternative policies, the results are consistent with our inferred optimum being an improvement over the type of policies one might have proposed a priori.

## 5   Discussion

The traditional experimental paradigm in psychology involves comparing a few alternative conditions by testing a large number of subjects in each condition. We've described a novel paradigm in which a large number of conditions are evaluated, each with only one or a few subjects. Our approach achieves an understanding of the functional relationship between conditions and performance, and it lends itself to discovering the conditions that attain optimal performance.

We've focused on the problem of optimizing instruction, but the method described here has broad applicability across issues in the behavioral sciences. For example, one might attempt to maximize a worker's motivation by manipulating rewards, task difficulty, or time pressure.

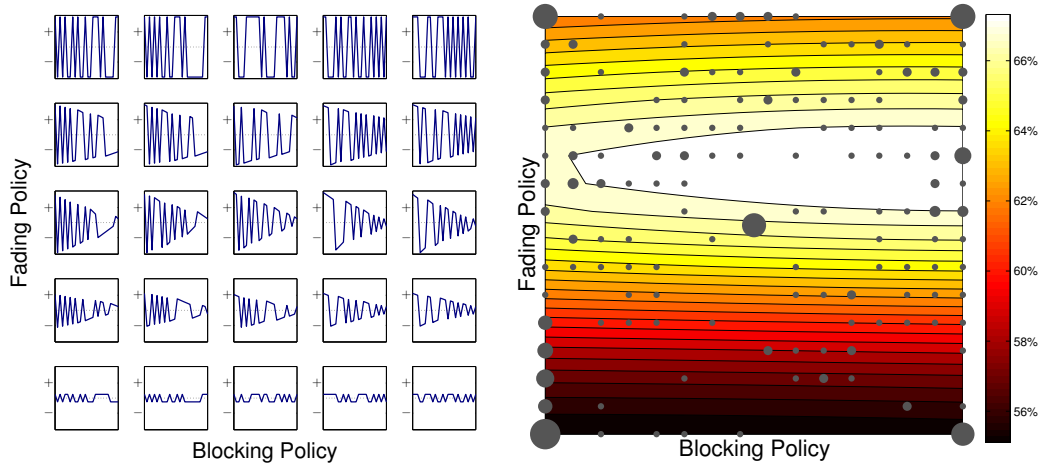

Figure 5: Experiment 2 (a) policy space and (b) policy performance function at 200 subjects

Motivation might be studied in an experimental context with voluntary time on task as a measure of intrinsic interest level.

Consider problems in a quite different domain, human vision. Optimization approaches might be used to determine optimal color combinations in a manner more efficient and feasible than exhaustive search (Schloss & Palmer, 2011). Also in the vision domain, one might search for optimal sequences and parameterizations of image transformations that would support complex visual tasks performed by experts (e.g., x-ray mammography screening) or ordinary visual tasks performed by the visually impaired.

From a more applied angle, A-B testing has become an extremely popular technique for fine tuning web site layout, marketing, and sales (Christian, 2012). With a large web population, two competing alternatives can quickly be evaluated. Our approach offers a more systematic alternative in which a space of alternatives can be explored efficiently, leading to discovery of solutions that might not have been conceived of as candidates a priori.

The present work did not address individual differences or high-dimensional policy spaces, but our framework can readily be extended. Individual differences can be accommodated via policies that are parameterized by individual variables (e.g., age, education level, performance on related tasks, recent performance on the present task). For example, one might adopt a fading policy in which the rate of fading depends in a parametric manner on a running average of performance. High dimensional spaces are in principle no challenge for GPR given a sensible distance metric. The challenge of high-dimensional spaces comes primarily from computational overhead in selecting the next policy to evaluate. However, this computational burden can be greatly relaxed by switching from a global optimization perspective to a local perspective: instead of considering candidate policies in the entire space, active selection might consider only policies in the neighborhood of previously explored policies.

**Acknowledgments**

This research was supported by NSF grants BCS-0339103 and BCS-720375 and by an NSF Graduate Research Fellowship to R. L. We thank Ron Kneusel and Ali Alzabarah for their invaluable assistance with IT support, and Ponesadat Mortazavi, Vanja Dukic, and Rosie Cowell for helpful discussions and advice on this work.

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
