[Reviews · NeurIPS 2013]

Submitted by Assigned_Reviewer_1

Paper 1279 – Optimizing Instructional Policies
In this paper, the authors adapt an optimization technique based on Gaussian process regression to select the parameters of experiments or teaching regime, which will optimize human performance. They evaluate their method in two behavioral experiments (one on the presentation rate of studied items and another on the ordering of examples while learning a novel concept), demonstrating that it is vastly more efficient than traditional methods in psychology for exploring a continuous space of conditions.

Note: I have revised my score to reflect the author feedback's assurance that the starting point of Experiment 1 wasn't the optimum. However, I still don't fully understand how what they wrote in the feedback connects to what they wrote in the paper. I implore the authors to make sure it is clear in the final version of their paper.

There is a lot to like in this paper. They use modern machine learning tools to solve a hot problem in cognitive science, optimal experimental/instructional design, and validate their method by replicating previous memory and concept learning experiments. Their adaptation of a previously developed optimization technique is natural and clever, albeit somewhat incremental. Besides for one potentially catastrophic issue issue with the first experiment (that I will describe below), the experiments do a good job of showcasing their method. Additionally, the paper is well written and the framing and discussion is very appropriate for a NIPS audience (though I have some concerns about the framing).
However, there are two serious issues with the paper that leave me very concerned.
1. Although Experiment 1 may seem at first to provide support for their method (as it shows quick convergence to the optimal presentation rate), after a closer reading the starting condition for the first subject is basically already at the optimum! “Subject 1’s training policy, $x_1,$ was set to the middle of the range.” The range of the policy space is [ln(.25), ln(5)]. This would make the midpoint 0.1116, which corresponds to a duration of 1.118 seconds. The reported optimum is 1.15 sec, which is extremely close to the starting value for the first subject (1.118 sec). If this is true (and I hope the authors can explain why this is not the case!), the new interpretation would at best be that the method does not move away from the optimum when it already is at it. This basically would invalidate the results of Experiment 1, and make the article difficult to support without its removal.
2. Although the paper is framed as being for constructing “instructional policies” and I am somewhat convinced that POMDP/RL methods are prohibitive to adapt for their situation, I really would have liked to have seen their method compared against at least one other computational method. It seems like there are several methods from optimal experimental design that would not be too difficult to adapt to this setting. In fact, I am not completely convinced that optimal experimental design is that different from finding optimal instructional policies (where instead the criterion is maximizing a function rather than discriminating between models viewed as functions). At this point there is a long tradition of optimal experimental design techniques that work over continuous spaces, some of which have already been applied to cognitive psychology. For example, Myung & Pitt (2009) Psych Rev. and other work from their group (including their 2009 NIPS paper), Kujala (2011) has a review on Bayesian adaptive estimation that discusses psychological applications, and Vul et al. (2009) in the journal Seeing and Perceiving (available for free from Vul’s website). The paper would have been much stronger if there was a direct comparison with at least some of these methods (even if they had to be adapted). In fact, although the framing of the paper is about instructional policies, the experiments and presented results do not seem to quite fit it (in the intelligent tutoring systems sense, which would be more of a discrete optimization problem). It seemed more about finding parameters/conditions to optimize human performance. However, I agree that Experiment 2 does a decent job of fitting the framing of the paper.
Minor comments:
Abstract: “… search this space to identify the optimum policy.” -> “… search this space to identify the optimal policy.”
The last paragraph of the introduction seems out of place. Perhaps an outline paragraph instead?
Second paragraph of Section 2: “The approach we will propose likewise requires specification of a constrained policy space, but does not make assumptions about the internal state of the learner or the temporal dynamics of learning.” No assumptions about the internal state of the learner seems a bit too strong. There are at least some minimal assumptions inherent in the generative model of the student. I agree they are not nearly as strong as some other methods though.
Third paragraph of Section 2: I would have liked to see a little more detail about why numerical approximation methods that account for noisy function evaluations would not work.
End of Section 2 - Section 2.1: Why outline the entire function if you are only interested in the maximum (and do not want any assumptions about the internal state of the learner)?
Section 2.2: you start to use $f_x$ before it is defined.
“In this paper, we give subjects questions with two equally probable choices…” How do you know the choices are precisely 0.5 in the minds of the subjects?
Discussion of limitations: what about discrete parameter spaces?
Section 3: If there are 30/d trials and d is continuous, what does a fractional trial mean?
Figure 3: Label units on x-axis (ms).
Experiment 2: Why did you have to discard objects with [2.75-3.25] graspability from the set?
Figure 4: I did not understand this plot (I printed it out on B&W, but even looking at the color, it does not really make sense to me).
Figure 5a: It could be nice to have large arrows across the subplots that provide an interpretation of what a small vs. medium vs. large fading/blocking policy means for the ordering of stimuli (small fading means close to far or random with respect to the boundary?)
Summary: A nice adaptation of machine learning methods to solve experimental design/"instructional policy" problems in cognitive psychology. However, there are some lingering serious concerns with Experiment 1 and the evaluation.

Submitted by Assigned_Reviewer_5

This paper represents a sophisticated attempt to apply Bayesian optimization to instructional policies. The authors present a theoretical framework and test it in two experiments.

QUALITY: I enjoyed the paper overall. Here are some specific commments:
- p. 3: what is a "reasonable subject budget"?
- figure 1: I don't think the cumulative density function is the most intuitive way to represent the posterior. Why not a confidence interval?
- It would be interesting ot know if different policy spaces can be better modeled by different covariance functions. For example, if the policy space is discrete, the squared-exponential covariance is probably not the best choice.
- I'm confused about the role of g in Eq. 1. Why should the probability that a random guess is correct be incorporated into the generative model in this way? Suppose that a subject never guessed randomly, even if g=0.5; then the binomial mean should be mu, but Eq. 1 says that the mean will be 0.5 + 0.5mu. I might simply be misunderstanding what's going on here, but if that's the case there needs to be a better explanation.
- What range of values are allowed for the parameters?
- For the beta-binomial likelihood, posterior inference over the latent performance function is intractable (not Gaussian, as the authors seem to suggest). What sort of approximation are the authors using?
- p. 6: I didn't get the point about the traditional experiment being less interpretable than the optimization procedure. Why is that?
- p. 7: interpreting the one significant t-test is a bit dubious, since you do not correct for multiple comparisons.
- The authors contrast their approach with model-based approaches using POMDPs and strong learning assumptions. It might be worth mentioning that the GP approach could also be used in tandem with model-based approaches. For example, a model could provide the mean function for the GP prior, in which case one could think of the GP model as capturing structure in the deviations from the parametric model.

CLARITY: The paper is clearly written.

ORIGINALITY: The work is original.

SIGNIFICANCE: This paper will be significant for psychology/education researchers interested in optimizing instruction, and will also be of interest to machine learning researchers as an application of GP optimization.

MINOR COMMENTS:
- p. 2: "agnostism" -> "agnosticism"
- p. 7: "issue discussed" -> "the issue discussed"
Summary: I think the framework is very promising on the whole, even though one of the experiments did not produce very compelling results.

Submitted by Assigned_Reviewer_7

This paper presents a new approach to optimizing instructional
policies based on searching a parameterized space of policies
and using GP function approximation to generalize from results
with a small number of subjects in many different conditions.

I liked the paper. I found the results of the experiments to be
intuitive and compelling. Technically the method seems sound and is
well presented. It would make a good NIPS paper because of its broad
relevance, not only to education but also to testing other systems
that have performance criteria one seeks to optimize with human users
(such as marketing and sales websites). The authors have a good point
that a method like the proposed one could be a good alternative to
more familiar A-B studies. If I have any reservations, it is that
there is nothing super exciting here. The idea of optimizing
educational policies makes sense, the technical approach seems
reasonable, and the results make sense. I'm glad someone has done
this; I'm glad the work exists. It's a solid paper. But I didn't
feel like I learned a great deal from it. If this were an education
or a cognitive science conference, I might feel differently, and say
this paper is potentially transformative for the field. But for NIPS,
well, it's not likely to be transformative, except perhaps for applied
researchers working in industry who might want to use this method to
optimize sales and marketing website design instead of traditional A-B
testing.

It would be interesting to know how general the results on example
sequences are, as far as teaching other kinds of concepts. One might
expect there to be some interaction between the nature of the concept
to be learned, the nature of the examples presented, and the optimal
teaching schedule. All the individual difference variables mentioned
at the end of the discussion would also be interesting to explore.
The local optimization approach that the authors describe at the end,
needed in order to optimize in higher dimensional policy spaces, would
be nice to see and perhaps might make for a more exciting contribution
to the NIPS audience.
Summary: A good solid paper using element of the NIPS machine learning toolkit to implement and test a novel exploratory-search approach to curriculum design. It would be of interest to both cog sci folks, but even more so probably to people working on website optimization.
Author Feedback

Author rebuttal: We thank all reviewers for their constructive suggestions for clarifying and extending our work. We address major reviewer concerns below.

R1 is concerned that our choice of the initial policy to be evaluated in Experiment 1 may have biased the experiment's outcome. It turns out that his reasonable assumption concerning our starting point was incorrect. The initial policy evaluated was actually 857 ms, about 200 ms from the optimum. (More on this value below; you can verify the value in Figure 3b.) The subject tested with that policy obtained an accuracy of 47/48. Although this clarification might address R1's concern, as a broader point we note that the choice of initial policy introduces a negligible bias for two reasons. First, the UCB active-selection strategy insists on exploring the untested regions of the space even if tested regions look promising. Second, model priors encode the expectation of significant intersubject variability (clearly present in the data, e.g., individual performance ranged from 70% to 100% at the optimum). Because our model searches for the _population_ optimum, no result from a single, noisy _individual_ heavily biases the GP posterior or the outcome of search. If search were driven by the first subject we tested, then the data point described above--a good score far from the optimum--might have _misled_ the search.

R1 also asked: if there are 30/d trials, how do we handle fractional trials? To ensure all conditions had the same training duration, we used only values of d such that 30/d was integer. Our initial policy, 857 ms, was the median of the range of admissible values of d. (Note that the GP framework is consistent with admission of only discrete values, even though the GP is defined over the continuous space.) We should have made this fact clear in the paper and will do so in a revision.

R1 implies that our approach is not appropriate for discrete parameter spaces. We disagree. GPs are suitable as long as a distance metric in the discrete space can be specified. Thus, our approach should prove useful for optimizing over binary decisions and/or rule orderings.

We thank R1 for pointing us to three cool papers on optimal experimental design. R1 contends that we should extend ("adapt") these methods and compare to our approach. We view these methods as complementary to, not as competitive with, our own work. Our work focuses on representing uncertainty over a space of policies and predicting performance under this uncertainty. In contrast, the Myung and Kujala papers are primarily concerned with active selection of the next experiment; Vul is about determining when to terminate experimentation. We have no commitment to the rather incidental decision we made to use the UCB approach common in the GP global optimization literature, and hope to consider the active-selection approaches of Myung and Kujala in subsequent work. We also see value in incorporating the Vul stopping criterion in subsequent work.

R5 notes we did not correct for multiple comparisons when we evaluated our hypothesized optimum against 3 alternatives. Our single significant result has p=.009. The Bonferroni correction involves multiplying this type I error by 3, yielding p=.027. We will revise the manuscript to reflect the corrected p value.

R5 asks why we claim that a traditional experiment is "less interpretable" than the outcome of our approximation method. Suppose we asked colleagues to estimate the policy function peak and/or shape based on the 5 data points of the traditional experiment (grey squares in Figure 3a). We expect that their interpretations would be highly variable, certainly more so than when shown the GP posterior mean (dashed line in Figure 3a).

R5 asked why we corrected for chance guessing by limiting the probabilities to [.5, 1], even though a particular subject might never guess randomly. The intuition behind this form is that the probability of guessing is closely tied to the efficacy of the policy a subject was trained on: subjects will always guess under bad policies (\mu = 0); and subjects will not need to guess under good policies (\mu = 1). Because our training procedures use veridical target labels, we did not entertain the possibility of policies that yield systematically incorrect responses.

We are not sure how R5 got the impression that we claim inference in our model is Gaussian, therefore tractable. We agree that posterior inference is intractable because of the chance-corrected beta-binomial likelihood. Section 2.2 states that we used a slice sampling technique for posterior inference. This recently developed MCMC technique is geared toward models like ours with Gaussian process priors & intractable posteriors. We used no analytic approximations.

R7 contends that NIPS may not be an appropriate venue for presenting state-of-the-art applications of machine learning methods. If so, NIPS should perhaps eliminate the cog sci, AI, comp vision, neurosci, and applications areas. We believe that the interaction between applications- and theory-focused researchers produces synergies. Our domain gives rise to unsolved problems in which we would hope to interest theoreticians. Two examples: (1) convergence of UCB search for global optimization with GPs has been shown--see Osborne reference in our bibliography---but no such proof exists for our model, whose latent function space and observation spaces are distinct and related via a nonlinear transformation. (2) We believe that optimization can be performed in higher dimensional spaces if we are willing to sacrifice global in favor of local optimization, but we would benefit from the enthusiasm of a theorist who could offer us concrete algorithmic approaches.